# Topology in time-reversal symmetric crystals

Jorrit Kruthoff,[1] Jan de Boer,[1] and Jasper van Wezel[1]

[1]*Institute for Theoretical Physics Amsterdam and Delta Institute for Theoretical Physics,
University of Amsterdam, Science Park 904, 1098 XH Amsterdam, The Netherlands*

The discovery of topological insulators has reformed modern materials science, promising to be a platform for tabletop relativistic physics, electronic transport without scattering, and stable quantum computation. Topological invariants are used to label distinct types of topological insulators. But it is not generally known how many or which invariants can exist in any given crystalline material. Using a new and efficient counting algorithm, we study the topological invariants that arise in time-reversal symmetric crystals. This results in a unified picture that explains the relations between all known topological invariants in these systems. It also predicts new topological phases and one entirely new topological invariant. We present explicitly the classification of all two-dimensional crystalline fermionic materials, and give a straightforward procedure for finding the analogous result in any three-dimensional structure. Our study represents a single, intuitive physical picture applicable to all topological invariants in real materials, with crystal symmetries.

Phase transitions in nature come in two types. The first is heralded by a change in symmetry, and includes for example the freezing of liquid water into ice, or the condensation of Cooper pairs into a superconducting state. A different sort of transition is traversed for example when changing from one conductivity plateau to another in the integer quantum Hall effect. This second type is related to the topology of the electronic wave function. Together, topology and symmetry determine the physical properties of any material, and they are the keystones in our modern understanding of phase transitions across all areas of physics. These two concepts are not, however, independent from one another. The much celebrated tenfold periodic table for example, lists the allowed topological phases depending on the types of time-reversal, particle-hole, and chiral symmetries present in a material [1–4]. The symmetries of the atomic lattice making up real crystals likewise restricts the number and types of topological phases that can emerge within them [5–15]. Combining symmetry and topology in such materials can give rise to exciting new features, like protected edge state circumventing the usual fermion doubling theorem, Fermi arcs, and isolated Weyl points [16]. It has yielded the discovery of weak topological invariants in three-dimensional time-reversal symmetric crystals [17], so-called bent Chern numbers [18], and translationally active topological states [14]. A systematic classification of all possible topological phases in the presences of a given crystal symmetry and dimensionality, however, has not yet been attempted in full generality.

The ideal approach, at least in principle, would be to use the rigorous mathematical tool of K-theory to find and index all topologically distinct phases of matter [19]. The challenge is that K-theoretic groups are notoriously hard to compute. As a result, there is no methodical mathematical structure that connects different types of known topological invariants, or guarantees that any list of topological invariants is complete in any but the simplest settings. Even the physical interpretation of what crystal features are represented by topological invariants varies wildly from one author to the next [20].

Here, we partially solve this problem for a large and experimentally relevant group of crystals. We present an algorithm for counting topologically distinct crystalline phases of fermionic matter with time-reversal symmetry (TRS), but broken particle-hole symmetry. That is, class AII in the tenfold periodic table [2, 4]. We also give an intuitive interpretation for the physical origin of all topological invariants encountered in these crystals. The presented algorithm augments our previous work on materials that have no symmetries other than those of their crystal structure (class A in the tenfold periodic table) [5]. In that restricted class, K-theories can be computed, and confirm the validity of our approach. The present work extends the intuitive counting procedure into the realm where results of K-theory are typically not available (class AII in the tenfold periodic table). Although this means the completeness of our classification cannot in general be rigorously proven in this class, confidence may be gained by the fact that it agrees with all results of K-theory that are available for systems in class AII. The approach described here thus provides for the first time a methodical algorithm for counting topological phases in time-reversal symmetric crystals. We use it to not only identify new crystal symmetries in which known invariant may arise, but also to suggest an entirely new topological invariant.

## Representation invariants

To find a way of counting the number of possible topological phases, we start by defining two insulating phases of matter to be topologically distinct (up to the addition of trivial bands), if smoothly deforming one into the other necessarily involves either closing the band gap around the Fermi level, or breaking a crystal symmetry [5]. These two conditions imply that symmetry eigenvalues can be used as a type of topological invariant, as shown for crystalline topological insulators in class A in [5]. Here, we review the arguments of that work, and generalise it to class AII.

| AZ class | type | p1 | p2 | pm | pg | cm | p2mm | p2mg | p2gg | c2mm | p4 | p4mm | p4gm | p3 | p3m1 | p31m | p6 | p6mm |
|---|---|---|---|---|---|---|---|---|---|---|---|---|---|---|---|---|---|---|
| AII | Representations | $\mathbf{Z}$ | $\mathbf{Z}$ | $\mathbf{Z}$ | $\mathbf{Z}$ | $\mathbf{Z}$ | $\mathbf{Z}$ | $\mathbf{Z}$ | $\mathbf{Z}$ | $\mathbf{Z}$ | $\mathbf{Z}^3$ | $\mathbf{Z}^3$ | $\mathbf{Z}^2$ | $\mathbf{Z}^4$ | $\mathbf{Z}^4$ | $\mathbf{Z}^3$ | $\mathbf{Z}^4$ | $\mathbf{Z}^4$ |
| | Torsion invariants | $\mathbf{Z}_2$ | $\mathbf{Z}_2^4$ | $\mathbf{Z}_2^2$ | $\mathbf{Z}_2$ | $\mathbf{Z}_2$ | $\mathbf{Z}_2^4$ | $\mathbf{Z}_2^2$ | $\mathbf{Z}_2^2$ | $\mathbf{Z}_2^3$ | $\mathbf{Z}_2^3$ | $\mathbf{Z}_2^3$ | $\mathbf{Z}_2^2$ | $\mathbf{Z}_2^3$ | $\mathbf{Z}_2^3$ | $\mathbf{Z}_2^3$ | $\mathbf{Z}_2^3$ | $\mathbf{Z}_2^3$ |

Table I. The classification of topologically distinct phases of two-dimensional crystalline matter in Altland-Zirnbauer class AII (i.e. having unbroken time-reversal symmetry, but broken particle-hole, chiral, and any other anti-commuting or anti-unitary symmetry). The topological invariants are either *torsion invariants* like the $FKM_2$ and line invariants, or *representation invariants* related to the transformation properties of the bands. The total classification is the (direct) sum of these two factors. The wallpaper groups in the first row are denoted in the Hermann-Mauguin notation [21].

Consider the example of a two-dimensional lattice with only four-fold rotation symmetry. In momentum space, the Brillouin zone has three high-symmetry points, $\Gamma$, $X$ and $M$. These momentum values are special because they are mapped onto themselves by at least some of the lattice symmetry operators. The wave functions making up the electronic bands at these high-symmetry points must be eigenstates of the symmetry operators. To be specific, $\Gamma$ and $M$ are invariant under the full four-fold rotation, whereas at $X$ there is only a two-fold rotational symmetry. Considering first the case with broken TRS, this means that at $\Gamma$ and $M$, each electronic band must have one of four possible eigenvalues, $\{\pm 1, \pm i\}$, while at $X$ only $\pm 1$ are allowed. We can now characterise a material with only four-fold rotation symmetry by listing the number of occupied bands for each eigenvalue at all of the high-symmetry points. This gives a list of ten numbers. In order for the assignment to be consistent throughout the Brillouin zone however, the total number of all occupied bands should be equal at all high-symmetry points. This gives two relations among the ten integers, resulting in a set of eight independent integers. These serve as eight topological invariants, because the only ways to change the number of bands with a given symmetry eigenvalue at a high-symmetry point, are to either break the symmetry, or take a band across the Fermi level. The list of eight numbers can thus not be changed without going through a topological phase transition [5]. We call these eight invariants *representation invariants*, because they specify the group representations of the lattice symmetry taken on by the electronic states.

If we include in our analysis the fact that electrons are spin$-\frac{1}{2}$ particles, the number of possible eigenvalues could change, because upon rotating the electron over $2\pi$, its wavefunction will be multiplied by $-1$, rather than 1. The possible eigenvalues of the symmetry operators are then contained in the fermionic part of the so-called double group. In the case of the four-fold rotational symmetry, there are still four different eigenvalues at both $\Gamma$ and $M$ and two at $X$, and the total number of integers acting as representation invariants is still eight.

Considering next the situation with time reversal symmetry (in this case with $T^2 = -1$), things do change. Each electronic state at momentum $k$ must now have a partner state with the same energy, but opposite spin, at $-k$. These two partner states necessarily come together into a single two-fold degenerate state at high symme-

try points. This is the celebrated Kramers degeneracy, and it is shown schematically in figure 1. Since one state in a Kramers pair is always related to a partner state by TRS, the transformations of a Kramers pair under symmetry operations now produce pairs of related eigenvalues. With only four-fold rotational symmetry, there is a single possible pair of eigenvalues at $X$, but two different allowed pairs at $\Gamma$ and $M$. Listing the number of occupied Kramers pairs in each representation thus gives five integers, which again are connected by two relations. In this case then, there are three independent representation invariants.

The counting of possible consistent sets of symmetry representations can be done for crystals in any dimension and with any crystal symmetry, for both broken and unbroken time-reversal symmetry (classes A, AI, AII, and AIII). In the Supplementary Material we give a detailed but straightforward algorithm for doing this consistently throughout the Brillouin zone for any crystalline material. The results for all time-reversal symmetric, two-dimensional, fermionic crystals are listed in table I. The corresponding result for any three-dimensional crystal can be easily found using the methods in the Supplementary Material. Notice that a generalisation of the arguments in [5] to class AI and AII was also given in [12, 22]. Here, we go beyond the results of those approaches by also considering the effect of crystal symmetries on topological invariants other than the space group representations themselves.[1]

## Torsion invariants

The representation labels are topological invariants, but by themselves they do not yet completely specify the band structure. Just like crystals with broken TRS may possess Chern numbers in addition to band labels, the representation invariants in crystals with unbroken TRS need to be supplemented with *torsion invariants*. These include the well-known Fu-Kane-Mele [23, 24], or $\mathbf{Z}_2$, invariants in two and three dimensions ($FKM_{2,3}$), as

---

[1] Notice that in class A, a relation between the representation invariants and the Chern number is known[6]. For classes AI and AII, however, it is not *a priori* clear whether such a relationship exists.

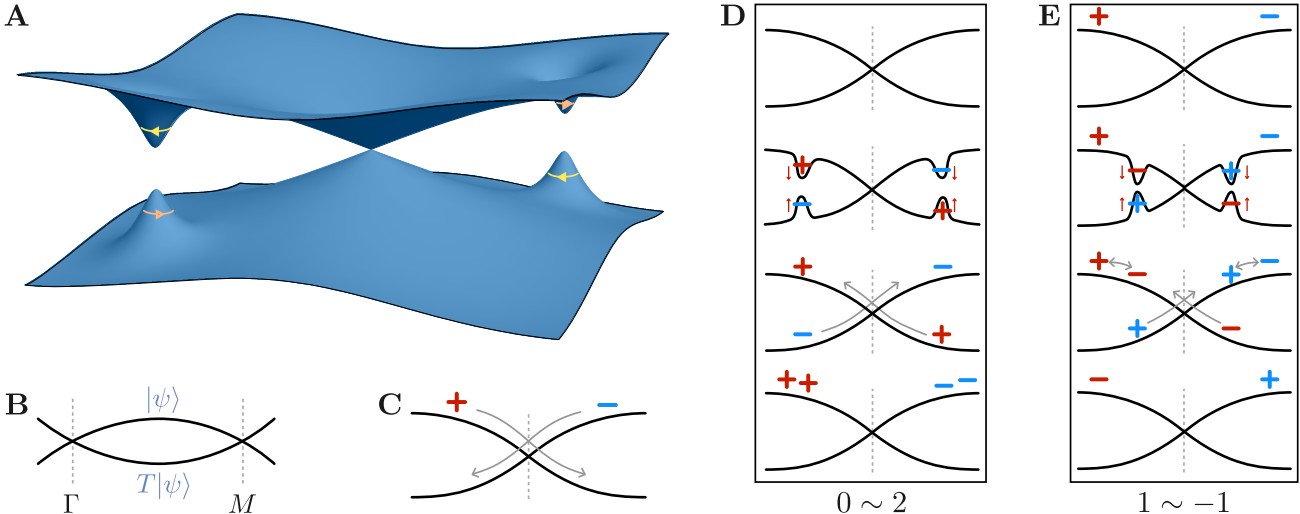

Figure 1. **a** The typical band structure of a Kramers' pair close to a high-symmetry point. Two bands related by the time-reversal operation necessarily come together into a degenerate Kramers pair at the time-reversal invariant momentum in the centre. Also shown schematically, is a band inversion which brings together states at points away from the high-symmetry momentum. This results in the formation of vortices in the Berry connection, indicated here by yellow and orange arrows. **b** A more schematic representation of two bands containing states $|\psi\rangle$ and $T|\psi\rangle$, which form Kramers pairs at two time-reversal invariant momenta, chosen here to be $\Gamma$ and $M$. **c** Vortices in the Berry connection, depicted by $+$ and $-$, can be moved throughout the Brillouin zone without annihilating. The color indicates the band to which the vortices belong. **d** An even number of vortices can be created by a band inversion within a set of states related by TRS. **e** Vortices can hop between partner bands using a band inversion to create two vortex anti-vortex pairs.

well as a generalisation of line invariants [10]. That crystal symmetries can be central in determining whether of not invariants other than the representation labels may arise in any given material is already known from the case with broken time-reversal symmetry. There, the famous Thouless-Kohmoto-Nightingale-den Nijs (TKNN) invariant, or total Chern number, is zero when reflection symmetries are present [6].

All torsion invariants are related to the presence of Berry curvature in some of the occupied electronic bands. To define a systematic procedure for identifying which torsion invariants are allowed to be non-zero in any time-reversal symmetric crystal, we interpret Chern numbers for individual bands as counting the number of vortices in its Berry connection. The generic procedure for creating such vortices is a continuous change in the Hamiltonian which closes the gap between two bands, takes them through each other, and again gaps any points of intersection. After this band inversion a vortex of one handedness resides in one of the bands, and one of the opposite handedness (an anti-vortex) in the other. Once formed, vortices can be moved throughout the Brillouin zone without closing any gaps, or breaking any symmetry, using non-topological changes in the Hamiltonian.

If the Hamiltonian is always time-reversal symmetric, then any change to an electronic state at momentum $\mathbf{k}$ is accompanied by an opposing change in the partner state at $-\mathbf{k}$. Vortices in TRS materials thus necessarily come in vortex anti-vortex pairs, as shown schematically

in figure 1. The pairs can be moved through the Brillouin zone, and even brought together at time-reversal invariant momenta, but they cannot annihilate there, due to the orthogonality of the electronic states within a Kramers pair. We give a more detailed analysis of this in the Supplementary Material.

Since vortices are created in pairs, the total vorticity, or total Chern number, within any pair of TRS-related bands is always zero. It is known however that vortices do not annihilate at high-symmetry points, because the (Berry) connection of the individual bands to the Kramers degenerate pair at the high-symmetry points does not mix the bulk time-reversed states [25]. This makes it possible to consider the Chern number of just one band within each pair, as proven rigorously in Ref. [25]. We have to keep in mind however, that a band inversion *within* the pair of TRS-related bands does not constitute a topological phase transition, as it does not close the gap at the Fermi level. As shown in figure 1, two vortices or anti-vortices can be created in each band this way, without changing the topological classification of the system. What cannot be done without going through a topological phase transition, is turning an even Chern number into an odd one. There is thus a $\mathbf{Z}_2$ invariant which can be expressed in terms of the Chern number $C$ of a single band as

$$
\begin{aligned}
\mathrm{FKM}_2 &= N \mod 2 \\
&= C \mod 2,
\end{aligned}
\tag{1}
$$

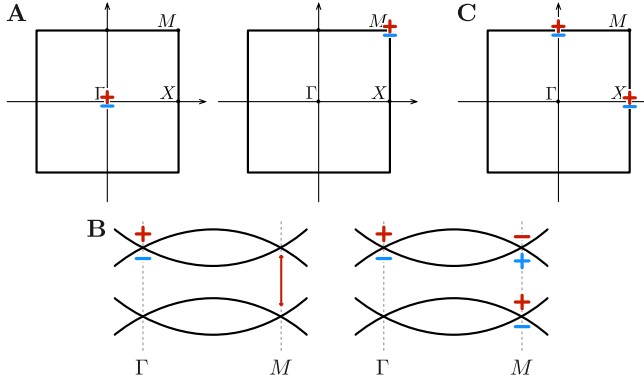

Figure 2. **a** Topologically non-trivial vortex configurations with $p4$ symmetry in class AII. **b** A band inversion involving a second, trivial, Kramers pair connects the configurations with a single vortex at $\Gamma$ to one with an $\text{FKM}_2$-trivial band with vortices at both $\Gamma$ and $M$, and one $\text{FKM}_2$-non-trivial band with only a vortex at $M$. Notice, however, that the final situation cannot be deformed into a band with a single vortex at $M$ and no vortices in the second band. That would require a change in the value of the new torsion invariant described in section C. **c** Vortex configuration with $p4$ symmetry in class AII in which the $\text{FKM}_2$ invariant is trivial, but the new invariant of section C is not.

with $N = N_+ - N_-$ the total vorticity, given by the difference in the numbers of vortices and anti-vortices. This is the Fu-Kane-Mele invariant for two-dimensional materials in class AII [17]. If multiple Kramers pairs are occupied, the corresponding $\text{FKM}_2$ invariants are summed.

A major advantage of the vortex picture of FKM invariants, is that the effects of crystal symmetry on its allowed values become much more transparent. In the lattice with only four-fold rotational symmetry for example, a vortex at some generic momentum $\mathbf{k}$ must always be accompanied by three other vortices at symmetry-related momenta. Such states have a topologically trivial FKM invariant ($\text{FKM}_2 = 0$) because $N$ is even. Topologically non-trivial states can be constructed by having a single vortex either at $\Gamma$ or $M$, whereas a vortex at $X$ again implies two vortices in the full Brillouin zone, and thus a trivial FKM invariant. All these configurations are shown schematically in figure 2, and described in more detail in the Supplementary Material, following a generalisation of the arguments given in [26] to non-trivial crystal symmetries.

In fact, a configuration with a single vortex at $\Gamma$ can be turned into a configuration with a single vortex at $M$ plus a band with trivial FKM invariant, if we allow for a second, trivial, Kramers pair to be present in the set of valence bands [8]. The two configurations are then connected by a band inversion, as shown in figure 2. As in the case without symmetries, the FKM invariant can thus take two possible values, signifying an even or odd number of vortices, without regard to where in the Brillouin zone the vortices occur.

## A. Line invariants

The identification of FKM invariants with vorticity, and the methodology of seeing how they are affected by lattice symmetries, works for all possible crystal structures in two and three dimensions, and is suited for class AI as well as AII. Additional features, however, may be identified if there are lines in the Brillouin zone that are mapped onto themselves by both TRS and a crystal symmetry, such as reflection, inversion, or two-fold rotation. On such lines, a one-dimensional topological invariant $\nu_1$, known as the line invariant or Lau-Brink-Ortix (LBO) invariant, can be defined [10].

The one-dimensional line invariants are in fact closely related to the vortices appearing in two dimensions. For example, in a crystal characterised only by a single reflection symmetry in the $x$ axis, the lines at $k_x = 0$ and $k_x = \pi$ are each mapped onto themselves by the reflection symmetry, and also by time-reversal. A line invariant can be defined on each of these lines, but they are related by the expression

$$\text{FKM}_2 = \nu_1^0 + \nu_1^\pi \mod 2. \tag{2}$$

The vortices in the Berry connection again provide an intuitive way to understand this. If $\text{FKM}_2 = 1$, there is one vortex at some momentum $\mathbf{k}$, and an anti-vortex in the time-reversed state with the same energy at $-\mathbf{k}$. Both of these must lie on the $k_x$ axis because of the reflection symmetry. Keeping in mind that reciprocal space is periodic owing to the translational symmetry of the atomic lattice, there are then two distinct ways the Berry connection between the vortices could behave. Examples of both are sketched in figure 3, which depicts a projection of the matrix-valued Berry connection onto the highest energy state. The connection either makes an odd number of complete windings along the line $k_x = 0$ and an even number along $k_x = \pi$, or the other way around. The field of Berry connections can be altered by gauge transformations and non-topological changes in the Hamiltonian. Since these do not affect the parities $\nu_1^0$ and $\nu_1^\pi$ of the number of windings along the two lines, however, the line invariants cannot be changed without going through a topological transition.

In the crystal with only a reflection symmetry, there are thus two ways for the FKM invariant to be non-trivial, depending on which of the two line invariants is non-trivial. Likewise, there are two ways for the FKM invariant to be trivial, having the line invariants either both zero, or both one. The latter case arises for example from a connection that winds the same way along all lines of constant $k_x$ but does not contain a vortex. The two independent torsion invariants in the crystal with only a reflection symmetry thus add a factor $\mathbf{Z}_2^2$ to its topological classification.

The heuristic arguments presented here in terms of vortices, are given a formal foundation in the Supplementary Material, where it is shown that the link between line invariants and the $\text{FKM}_2$ invariant, arising from vortices

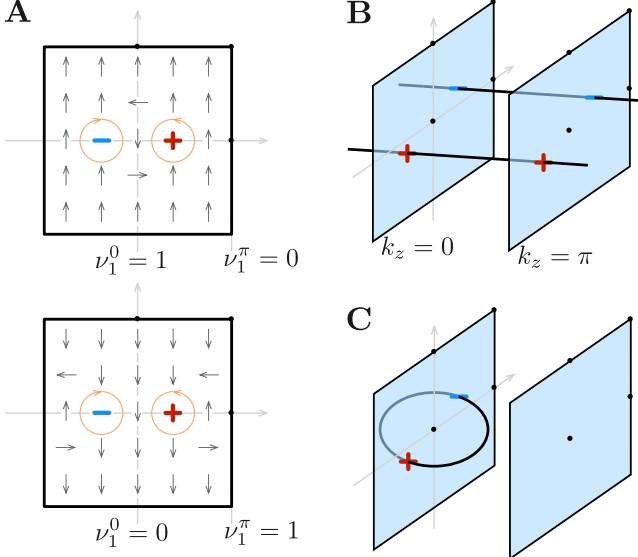

A

$\nu_1^0 = 1 \qquad \nu_1^\pi = 0$

$\nu_1^0 = 0 \qquad \nu_1^\pi = 1$

B

$k_z = 0 \qquad k_z = \pi$

C

Figure 3. **a** Sketch of the Berry connection projected onto the highest energy state within a Kramers pair. A vortex and anti-vortex pair can arise in two topologically distinct ways within a Berry connection vector field that is continuous on the Brillouin zone torus. **b** Sketch of vortex lines extending across the bulk of a three-dimensional Brillouin zone with trivial FKM$_3$ invariant. **c** Sketch of a vortex line extending into the bulk of a three-dimensional Brillouin zone, and closing onto itself. This situation is described by a non-trivial FKM$_3$ invariant.

in the Berry connection, holds in general. To fully classify topological insulators both of the torsion invariants, as well as the relations between them, need to be consistently taken into account. This can be done for any crystal symmetry in two and three dimensions using the analysis detailed in the Supplementary Material.

### B. Integer spin

For spinless electrons, in class AI of the ten-fold periodic table, one may not expect the FKM and line invariants to play a role [2]. Combining spatial symmetries with TRS, however, can cause bands of spinless electrons to mimic the structure of a Kramers pair, by forcing bands with complex eigenvalues of symmetry operations to necessarily become degenerate at high-symmetry points. In these cases, non-trivial torsion invariant are again allowed [7]. Whether or not further, different, types of torsion invariants can arise in this class, is a question we leave for future investigation.

### C. A new invariant

The combination of line and FKM invariants constitutes all known torsion invariants in time-reversal sym-

metric crystals. This, however, cannot be the full picture. Consider, for example, the crystal with only two-fold rotational symmetry. There are many lines in the Brillouin zone that can be mapped onto themselves by both TRS and the two-fold rotation. Most of these lines can be smoothly deformed into one another, and it suffices to define line invariants on the $k_x = 0, \pi$ and $k_y = 0, \pi$ lines. These are again related to each other by the FKM$_2$ invariant, giving a total of three independent torsion invariants.

A possible configuration with all invariants equal to zero would be to have no vortices present in the band structure at all. Another possible configuration with the same values for all invariants would be to have vortices present at all high-symmetry points. Because of the rotational symmetry, however, vortices cannot be spread out away from the high-symmetry points by any deformation of the Hamiltonian. That is, all Berry curvature is always concentrated in delta-peaks at the high-symmetry points, as shown in more detail in the Supplementary Information. But this means that the situation with four vortices can only be deformed into the situation without vortices if either the gap is closed or the symmetry broken. These two phases must thus be considered topologically distinct, and there must exist an additional $\mathbf{Z}_2$ or torsion invariant distinguishing them.

In fact, it is easily seen that every combination of values of for the two line invariants and one FKM invariant can be realised with precisely two distinct configurations of vortices on the high-symmetry points. Again, these can never be smoothly deformed into each other, and should be distinguished by the new torsion invariant. Additional evidence for the existence of the new invariant can be found in two places. First of all, it is known that in certain cases a band structure with an odd total number of vortices in all valence bands at the $\Gamma$ point has distinct physical properties from a band structure with an odd number of vortices at $M$, even if all line and FKM invariants are the same [14, 27]. This difference is manifested when a topological defect is introduced into the crystal, which will be either charged or not, depending on the configuration of vortices [27]. The topological defect in such cases may thus be seen as indicator for the new invariant.

Furthermore, in the specific case of a crystal with only two-fold rotational symmetry, the K-theory in the presence of time reversal symmetry may be explicitly computed, as discussed in more detail in the Supplementary Information. This shows that in this specific case, the Brillouin zone hosts two invariants at its edges, and two invariants in its bulk. These correspond directly to the two line invariants, the one FKM invariant, and the one new invariant found by counting vortices. Notice that although K-theory calculations in the presence of TRS are very challenging in all but this simplest case, counting vortices in topologically distinct situations as suggested in the current approach is always straightforward.

In each of the situations with equal line and FKM in-

variants but different vortex configurations, the topologically distinct phases can be distinguished by finding out whether or not a vortex is present at $\Gamma$. The new invariant can thus be determined by calculating the Berry curvature of a single Kramers pair partner in a small region encircling the $\Gamma$ point. As is shown in more detail in the Supplementary Information, this procedure is guaranteed to be well-defined, because the rotational symmetry forbids the spreading of Berry curvature away from high-symmetry points.

An especially interesting situation to consider in the light of this new invariant, is that of a crystal with three-fold symmetry. In that case, there is a TRS point at $\Gamma$ with rotational symmetry, a TRS point at $M$ without any point group symmetry, and a point at $K$ that is invariant under rotations, but not under TRS. Looking at the allowed representations at $\Gamma$, there is one real representation that allows for three vortices (or equivalently, a single charge-three vortex) to be formed there. These vortices can be moved to $M$ or $K$ by transformations of the Hamiltonian that do not close the gap or break the lattice symmetry. However, there is also a complex representation at $\Gamma$, which allows for a single (charge-one) vortex to be formed there. This single vortex cannot be moved away from $\Gamma$, because of the rotational symmetry. It can also not be transformed into a situation with three vortices without going through a topological phase transition. A similar charge-one vortex may also exist at $K$, accompanied by an anti-vortex at $-K$, and again such a vortex cannot be moved away from the high-symmetry point. The parity of the numbers of charge-three vortices anywhere in the Brillouin zone, and charge-one vortices at $\Gamma$ and at $K$, are therefore three independent torsion invariants. Notice that in this case, the representations of the bands at $\Gamma$ in fact determine which $Z_2$ invariants are allowed. This is reminiscent of the way that rotational symmetries of the lattice may be used to determined the Chern number of class-A materials modulo the order of the rotation [28].

Combining the list of allowed torsion invariants with that of representation invariants, table I presents the full classification of spin-full electrons in two-dimensional crystals with time-reversal symmetry. The total classification is the direct sum of the representation and torsion invariants. This does not exclude the possible existence of relations amongst them. In fact, we already encountered such relations between representation invariants and Chern numbers in class A, as well as for example for materials with $p3$ symmetry in class AII. As far as the counting of topological invariants is concerned, however, the total classification is given by the sum of invariants. The same algorithm can be used to straightforwardly compute the analogous table for three-dimensional crystals and layer groups, keeping in mind there may be additional torsion invariants in higher dimensions.

## D.   Three dimensions

In three dimensions, the analysis of symmetry eigenvalues and the corresponding representation invariants is completely analogous to that in two dimensions. The torsion invariants on the other hand, feature an additional entry special to three dimensions, the $\mathrm{FKM}_3$ invariant. To understand this invariant in terms of the vortices in the Berry connection, consider the planes $k_z = 0$ and $k_z = \pi$, which are mapped onto themselves by the time-reversal operation. On these planes, two-dimensional $\mathrm{FKM}_2$ invariants may be defined. Much like line invariants are related to $\mathrm{FKM}_2$, the invariants of the two planes are related to $\mathrm{FKM}_3$ by the expression

$$\mathrm{FKM}_3 = \mathrm{FKM}_2^0 + \mathrm{FKM}_2^\pi \mod 2. \tag{3}$$

An intuitive understanding can again be found using vortices in the Berry connection. A single vortex and anti-vortex on for example the plane $k_z = 0$ can be extended into the third direction as a vortex line, or flux tube. If the vortex line extends all the way to the plane $k_z = \pi$, both planes have non-trivial $\mathrm{FKM}_2$ invariants. On the other hand, if the line closes onto itself and forms a vortex loop, the $\mathrm{FKM}_2$ invariant at $k_z = \pi$ will be trivial, and there will be a non-trivial $\mathrm{FKM}_3$ invariant in the bulk of the Brillouin zone. This situation is shown schematically in figure 3. Notice that a single $\mathrm{FKM}_3$ invariant may connect multiple parallel planes on which $\mathrm{FKM}_2$ invariants can be defined. Incorporating the effect of crystal symmetry on whether or not $\mathrm{FKM}_3$ invariants are allowed is a matter of understanding the effects it has on vortex lines. When inversion symmetry is present, it is known that $\mathrm{FKM}_3$ can be computed using the inversion eigenvalues [29], and is therefore absorbed in the representation invariants. A more detailed derivation of these heuristic arguments is given in the Supplementary Material.

An interesting example of a three-dimensional crystal, is one with space group $P2/m$ (nr. 10). Such a crystal has inversion symmetry, and a two-fold rotation symmetry around the $k_z$-axis. The representation invariants can be straightforwardly identified both for spinless and spin$-\frac{1}{2}$ particles, as is done in the Supplementary Material. Spinless particles (class AI) do not have torsion invariants for this crystal symmetry, due to the lack of a Kramers pair structure. In class AII, the torsion invariants on $k_z = 0, \pi$ planes cannot be non-trivial, because the inversion symmetry forbids single vortices even at high-symmetry points. Since the values of both the line invariants and new invariants are related to the presence of vortices at high-symmetry points, these too must be trivial. Moreover, due to (3), $\mathrm{FKM}_3$ is also zero. We thus find no torsion invariants in this crystal. Notice that this implies the $\mathbf{Z}_2$ invariant in for example [29], is in our description absorbed into the representation invariants.

## E. Discussion

Interpreting topology in insulators to stem from a combination of representation invariants imposed by the symmetries of the atomic lattice, and torsion invariants that can always be interpreted as coming from vortices in the Berry connection, provides a straightforward way of counting the number of topological invariants needed to classify time-reversal symmetric crystalline materials in any dimension up to three. The picture is especially powerful, however, in relating topological invariants associated with different dimensions, and gives an intuitive, unified picture of how these are influenced by the presence of both lattice symmetries and each other. In doing so, it reveals the necessary existence of a hitherto unknown topological invariant complementing the line and FKM invariants.

As mentioned in passing in the introduction, we consider two bands to be distinct under smooth deformations up to the addition of trivial bands. More explicitly, this implies that two band structures are topologically equivalent when they can be made to be equal upon adding topologically trivial sets of bands. In the present context, and in accord with K-theory, trivial band structures are defined to be particle-hole symmetric pairs of bands. To be precise then, we really consider the combined topological invariant of all bands below a gap in the spectrum at any energy (not necessarily at the Fermi level), and consider the trivial set of bands to be a pair with equal topological indices in which one is occupied and one unoccupied. This definition reflects the fact that negative integers may appear in the K-theory, and in our classification, corresponding to bands of holes, rather than electrons. This necessity of including the concept of negative integers in the definition of equivalence is a direct consequence of the fact that the elements in K-theory are difference classes, which necessitates the existence of a trivial element.

Extrapolating the bulk-boundary correspondence for known topological insulators to the range of new topological phases identified here, suggests that new boundary modes may be associated with at least some of the new invariants characterising these materials. The existence and properties of these new modes will be an interesting avenue for future research. Likewise, the intuitive arguments presented here are given a solid mathematical foundation in the Supplementary Material, which ensures a consistent counting of torsion and representation invariants for all crystalline, time-reversal invariant materials in class AII up to three dimensions. We cannot yet, however, give an explicit mathematical proof that these invariants exhausts all possible topological quantum numbers. To do that, a comparison to a purely K-theoretic analysis would be required, which we hope will become available in the near future.

*Acknowledgement*—We would like to thank Adrian Po, Aaron Royer, and Jan Zaanen for several helpful discussions. We are also indebted to Luuk Stehouwer for bringing the possible existence of a new invariant to our attention. JK is supported by the Delta ITP consortium, a program of the Netherlands Organisation for Scientific Research (NWO) that is funded by the Dutch Ministry of Education, Culture and Science (OCW). JvW acknowledges support from a VIDI grant financed by the Netherlands Organization for Scientific Research (NWO).

---

## I. SUPPLEMENTARY INFORMATION

In the sections below, we start out by giving a brief summary of the representation theory of space groups in the presence of time-reversal symmetry. We will only cover those things that are related directly to the main text. Other subjects and more in-depth discussions can be found for example in [30, 31].

In the remainder we then focus on the formal description of the torsion invariants introduced in the main text, in terms of vortices and line invariants in band structures. The discussion then continues with a formal description of the FKM invariant in terms of transition matrices and the topologically non-trivial classes that emerge as a result of combined TRS and lattice symmetries. We then discuss the relations between representation and torsion invariants, as well the relations between different types of torsion invariants. We end with a brief discussion of the K-theoretical calculation that can be used to show the necessary existence of a new torsion invariant.

## II. TIME-REVERSAL SYMMETRY AND SPACE GROUPS: MAGNETIC SPACE GROUPS

### A. Types of magnetic space groups

Magnetic or non-magnetic materials, for example in a magnetically disordered phase or ferromagnet, may posses anti-unitary symmetries. This means that the original space group of the lattice $G$ needs to be enlarged by inclusion of an anti-unitary operator $a$. In general, the enlarged group, called the magnetic space group is then

$$M = G \oplus aG. \tag{4}$$

Depending on what $a$ is, there are three types of magnetic space groups (we exclude the trivial option in which $a$ is not present). When $a = \Theta$, the time-reversal operator, $M$ is called a type-II Shubnikov space group. Notice that in this magnetic space group, TRS commutes with all elements of $G$ and the crystal is non-magnetic.

If a system is magnetic, it could still be invariant under an anti-unitary operator, but not under $\Theta$ alone. TRS should then be accompanied by either a rotation or a reflection, allowing the system to be invariant under a type-III Shubnikov space group,

$$M = H \oplus aH, \tag{5}$$

where $H$ is a index-two subgroup of $G$ (the original space group). The anti-unitary symmetry is now $a = R\Theta$, where $R$ is a point group operation of $G$ such that $G = H \oplus RH$.

There is also a third kind of magnetic space group, the type-IV Shubnikov space group. In this case, the time-reversal operator is accompanied by a translation, $\mathbf{t}_0$, so that:

$$M = G \oplus \Theta\{E|\mathbf{t}_0\}G. \qquad (6)$$

From here on, we will focus on the case in which time-reversal symmetry is really a symmetry of the system itself, and consider only type-II Shubnikov space groups. The other magnetic spaces groups can be studied in a similar way [13].

## B. Representation theory

To find the representation theory of magnetic space groups, we first focus on the action of the time-reversal operator $\Theta$ on the bands, and let it act on $\rho(g) |\psi_n\rangle$ (where $n$ is the band index), ignoring any momentum dependence for now. The operator $\rho(g)$ is any unitary operator corresponding to an element $g$ of $G$ that acts on the states according to some representation $\rho$ of the space group $G$. Now,

$$\mathcal{T}\rho(g) |\psi_n\rangle = \rho^*(g)\mathcal{T} |\psi_l\rangle \qquad (7)$$

with $\mathcal{T}$ a representation of $\Theta$. Representations of the magnetic space group need to satisfy this relation. These representations are called *co-representations* [32]. It is straightforward to show that these representations have the following properties. First of all, the time-reversed representation $\hat{D}(g)$ of some element $g$ is equivalent to the complex conjugated representation of $g$, i.e.

$$\hat{D}(g) = D(g)^* \qquad (8)$$

Second, in addition to adding symmetry elements, time-reversal symmetry can also enhance the state space. A prime example is the emergence of Kramers pairs. These are formed because $|\psi\rangle$ and its time-reversed partner can be guaranteed to be orthogonal, causing the matrix representations to be twice their original size. Expressing these new representations in terms of the original representations of the space group allows us to directly apply the algorithms introduced in Refs. 5 and 12 for assigning symmetry labels to bands.

Consider a general magnetic group $M = G \oplus aG$, and suppose that $g$ is an element of $G$, the space group. Then in the basis $\{|\psi\rangle, a|\psi\rangle\}$, the representation of $g$ is

$$D(g) = \begin{pmatrix} \rho(g) & 0 \\ 0 & \rho^*(a^{-1}ga) \end{pmatrix} \qquad (9)$$

However, for the other half of the elements of $M$, elements of the form $b = ag \in aG$, the representation looks like

$$D(b) = \begin{pmatrix} 0 & \rho(ba) \\ \rho^*(a^{-1}b) & 0 \end{pmatrix} \qquad (10)$$

These representations are irreducible in the sense of Ref. 30. Intuitively, TRS is understood as a symmetry that can cause bands to stick together to form Kramers pairs. This can happen in three ways. Either nothing happens, or complex conjugate irreducible representations stick together, or the bands just become doubled. In detail these three cases are:

a) In this case $\rho(g)$ is unitarily equivalent to $\rho^*(a^{-1}ga)$, i.e. $\rho(g) = N\rho^*(a^{-1}ga)N^{-1}$. Where $N$ satisfies $NN^* = +\rho(a^2)$, then $D(g) = \rho(g)$ and $D(b) = \pm\rho(g)N$.

b) In this case $\rho(g)$ is unitarily equivalent to $\rho^*(a^{-1}ga)$, i.e. $\rho(g) = N\rho^*(a^{-1}ga)N^{-1}$. Where $N$ satisfies $NN^* = -\rho(a^2)$, then

$$D(g) = \begin{pmatrix} \rho(g) & 0 \\ 0 & \rho(g) \end{pmatrix} \quad D(b) = \begin{pmatrix} 0 & -\rho(g)N \\ \rho(g)N & 0 \end{pmatrix} \qquad (11)$$

c) In this case $\rho(g)$ is not unitarily equivalent to $\rho^*(a^{-1}ga) = \bar{\rho}(g)$. The magnetic space group representations are then given by

$$D(g) = \begin{pmatrix} \rho(g) & 0 \\ 0 & \bar{\rho}(g) \end{pmatrix} \quad D(b) = \begin{pmatrix} 0 & \rho(ga^2) \\ \bar{\rho}(g) & 0 \end{pmatrix} \qquad (12)$$

To determine whether we are dealing with type (a), (b) or (c) upon inclusion of TRS we use a test deviced by Herring in 1937 based on the Frobenius-Schur indicator. Given a (projective) irreducible representation $\rho_{\mathbf{k}}$ of the little co-group at $\mathbf{k}$, we can write this test as

$$I(\rho_{\mathbf{k}}) = \frac{1}{\#S_i} \sum_{S_i} e^{-i(\mathbf{k}+S_i^{-1}\mathbf{k})\cdot\mathbf{w}_i} \rho_{\mathbf{k}}(g_i^2)$$

$$= \frac{1}{\#S_i} \sum_{S_i} e^{-i\mathbf{g}\cdot\tau_i} \rho_{\mathbf{k}}(g_i^2). \qquad (13)$$

where the sum is over those $S_i = \{g_i|\tau_i\}$ such that $g_i \cdot \mathbf{k} = -\mathbf{k}$ modulo a reciprocal lattice vector $\mathbf{g}$. The fractional translation associated to $S_i$ is denoted by $\tau_i$. Thus when $\mathbf{k} \equiv -\mathbf{k} + \mathbf{g}$ (i.e. at high-symmetry points which are also TRS invariant points), we sum over all elements of the little co-group of $\mathbf{k}$, $G^{\mathbf{k}}$. The value of $I(\rho_{\mathbf{k}})$ determines whether the irreducible representation $D_{\mathbf{k}}$ arising from $\rho_{\mathbf{k}}$ by adding time-reversal symmetry, is of type (a), (b) or (c). The assignment follows from:

$$I(\rho_{\mathbf{k}}) = \begin{cases} \gamma & \text{case (a)} \\ -\gamma & \text{case (b)} \\ 0 & \text{case (c)} \end{cases} . \qquad (14)$$

with $\gamma$ being the sign of the square of the time-reversal operator $\Theta$. This test can be used for any of the three Shubnikov space groups, because one can write a general anti-unitary element as a space group element times $\Theta$.

## C.  TRS degeneracies

Now that we know where degeneracies occur, we need to compute the irreducible representations that stick together. For cases where $I = \pm 1$, this is trivial, but for $I = 0$ it is not. Let us assume that we are at a high-symmetry point which has $G^{\mathbf{k}}$ as its little co-group and that $I = 0$ for some, possibly projective, irreducible representations of $G^{\mathbf{k}}$. Also we assume the magnetic little co-group is given by $M^{\mathbf{k}} = G^{\mathbf{k}} \oplus a G^{\mathbf{k}}$, with $a = \Theta a_0$. It is important to note that $a_0$ is not part of $G^{\mathbf{k}}$ and so multiplication is done within the full point group. The TRS reversed representation is given by

$$\bar{\rho}(S) = \rho(a_0^{-1} S a_0)^*, \tag{15}$$

where $S = \{g|\tau\}$ with $\tau$ a fractional translation and $a_0 = \{g_0|0\}$. This can be rewritten using

$$\begin{aligned} a_0^{-1} S a_0 &= \{g_0^{-1}|0\}\{g|\tau\}\{g_0|0\} \\ &= \{e|g_0^{-1}\tau - \tau\}\{g_0^{-1}g g_0|\tau\}, \end{aligned} \tag{16}$$

where $e$ is the identity element. Thus (15) becomes

$$\bar{\rho}(S) = \exp(i\mathbf{k} \cdot (g_0^{-1}\tau - \tau))\rho(\{g_0^{-1}g g_0|\tau\})^*. \tag{17}$$

Now there are two possible situations. First we could have $a_0 = \{e|0\}$ ($M^{\mathbf{k}}$ is a type-II Shubnikov space group), in which case

$$\bar{\rho}(S) = \rho(\{g|\tau\})^*. \tag{18}$$

This situation occurs when $\mathbf{k}$ is also a TRS invariant point. The other option is $g_0 \neq e$ with $g_0 \cdot k = -k$ and so

$$\bar{\rho}(S) = \exp(-i\mathbf{k} \cdot \tau)\rho(g_0^{-1}g g_0)^* \tag{19}$$

where the product $g_0^{-1}g g_0$ should be calculated in the full point group, which might be realised projectively. For example when $G^{\mathbf{k}}$ consists of a single glide plane (f.e. $p2mg$ or $p2gg$) and $a_0$ is a reflection in the $k_x$ axis, then the multiplication should be done in the central extension, i.e. in the quaternion group. In this group the reflections anti-commute.

## III.  SMEARING BERRY CURVATURE

In a class A topological insulator, the bands are generically non-degenerate. In that case we can move two single bands close to each other at some point in $k$ space and let them invert. This band inversion is, generically, responsible for non-trivial topology in the valence bands. In two dimensions, close to the region in $k$ space where the bands meet, the Hamiltonian takes the form

$$H = k_x \sigma_x + k_y \sigma_y + m \sigma_z \tag{20}$$

with $m$ a small mass between the two bands, one belonging to the valance bands, the other to the conduction bands. The Berry curvature of the band belonging to the valance bands is then

$$F_{k_x k_y} = \frac{-im}{2} \frac{1}{(k^2 + m^2)^{3/2}}, \tag{21}$$

where we ignore regularisation issues for the moment, because they will not be important for our argument. One sees that when $m \to 0$, the curvature localises at $k_x = 0 = k_y$ and that increasing the mass can be viewed as smearing the curvature over a small region around $k_x = 0 = k_y$. When we add rotation symmetry in class A such smearing is still allowed.

However, if we move to class AII, bands at time-reversal invariant points become degenerate Kramers pairs. Such degeneracies are present in both the valance and conduction bands at energies away from the Fermi energy. The Hamiltonian in (20) can also be used to describes a Kramers pair near a high-symmetry point, but only if $m = 0$, as any nonzero $m$ will destroy time-reversal symmetry. The argument that we used above to smear a single vortex in a non-degenerate band, can thus not be used to smear the Berry curvature contained in a vortex anti-vortex pair localised at a time-reversal symmetric high-symmetry point. In other words, if curvature is introduced at time-reversal invariant points, it necessarily remains localised there. Notice, however, that the Hamiltonian in (20) only describes the band structure near time-reversal invariant points. At generic points in the bulk of the BZ, the bands are non-degenerate and vortices can be created by band inversions involving only a single valence band. Berry curvature at such points can be smeared as usual.

Let us now consider the effect of adding spatial symmetries. A rotation symmetry will force vortices created at generic point to come in multiplicities equal to the order of the rotation and hence for the even-fold rotation groups only an even number of vortices will be created signalling trivial topology. As explained in the main text, non-trivial topolog thus requires vortices to be formed at time-reversal invariant points. These can subsequently not be smeared and are stuck at those high-symmetry points. In the presence of reflection symmetries vortices are similarly stuck on high-symmetry lines.

### A.  Stuck vortices and their invariants

Berry curvature localised on high-symmetry lines can be detected by computing the one-dimensional line, or LBO, invariant [10]. The vortices stuck to time-reversal invariant points also constitute invariants and to detect them one can simply integrate the curvature of a single band within the Kramers' pair over a small region around such points. Isolating a single band within the Kramers' pair to compute the invariant this way can be done in exact analogy to how the FKM invariant is computed from the Berry curvature over the whole BZ [25]. Since the curvature is contained within a $\delta$-function localised

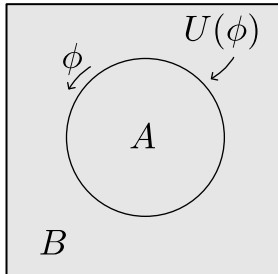

Figure 4. The BZ divided into two patches, $A$ and $B$, with a transition function $U(\phi)$ in between. The coordinate $\phi$ parametrizes the overlapping circle between $A$ and $B$.

at time-reversal invariant points, any surface round the vortex may be considered, as long it encloses only a single time-reversal invariant point.

## IV. TRANSITION FUNCTIONS

Next, we turn to a simple way of understanding the torsion invariants introduced in the main text in terms of transition functions. Such functions are necessary to globally specify the vector bundle of states above the BZ, and can be straightforwardly constructed. See Ref. 33 for a clear exposition of transition functions and vector bundles. We will also comment on line invariants in terms of transition functions, and consider their relation to the $FKM_2$ invariant. In the remaining we will mostly focus on systems with two bands, and generalise the approach presented in Refs. 26 and 34. Notice that the following analysis is only a local. We believe that there generally are global constraints, but we cannot prove that the analysis with the constraints is equivalent to a K-theory computation.

The possibility of having an $FKM_2$ invariant in class AII signals the fact that it may be impossible to globally define a basis for the two bands in a Kramers pair. To show that this fact is related to the parity of the Chern number, one defines two patches $A$ and $B$ in the BZ. To be precise, the patches should be topologically trivial, which means that on the torus, two patches are not enough. The reason that we still consider only two patches, is that the $FKM_2$ invariant is most easily identified in the equivariant K-theory of the sphere. The fixed points of the point group outside, say $\Gamma$, can be collapsed to a single point, leaving only two fixed points on the sphere, and this justifies the use of just two patches.

In one patch one can define a consistent basis for the Kramers doublet, but the basis might not be the same in the other patch. The change in basis between the two patches is encoded in a transition function, as shown schematically in Fig. 4. A transition function for a topologically trivial system would be

$$U(\phi) = \begin{pmatrix} e^{2i\phi} & 0 \\ 0 & e^{-2i\phi} \end{pmatrix}, \qquad (22)$$

whereas in the non-trivial case it could be given by

$$U'(\phi) = \begin{pmatrix} 0 & e^{i\phi} \\ e^{-i\phi} & 0 \end{pmatrix}. \qquad (23)$$

To find these transition matrices, one simply imposes TRS on a general unitary two by two matrix: $i\sigma_y \bar{U}(\phi)(-i\sigma_y) = U(\phi + \pi)$. The matrix $U$ is a $U(2)$ matrix which can be decomposed uniquely as $U(\phi) = e^{i\theta(\phi)}g(\phi)$, with $g \in SU(2)$. There are two classes of $U's$ that satisfy the TRS condition. Either we have $\theta(\phi) = -\theta(\phi + \pi)$ and $g(\phi) = g(\phi + \pi)$ or $\theta(\phi) = -\theta(\phi + \pi) + \pi$ and $g(\phi) = -g(\phi + \pi)$. The $\pm$ sign for $g$ are the only possibilities consistent with $g$ being an $SU(2)$ matrix. This sign really comes from the $U(1)$ part of $U(2)$ and contains all the topology. The $SU(2)$ part has, on the level of the fundamental group, no topology, i.e. $\pi_1(SU(2)) = \emptyset$ and so within each of the two classes, the transition matrix can be deformed at will, as long as the condition on the $U(1)$ factor is not violated. Of course, one can also turn the logic around and argue that due to the two choices on the $U(1)$ part of $U$, the $SU(2)$ part needs to satisfy certain periodicity conditions.

As for the transition matrices given above, one quickly checks that in the trivial case, the bands have Chern number $\pm 2$, whereas in the non-trivial case they are $\pm 1$. In fact, as was shown in Refs. 26 and 34, only the parity of the Chern number in each band is a topological invariant.

Let us now see what changes to the transition functions as we add rotation symmetry. The action of rotation symmetry on the transition function is encoded in the irreducible representations of the double group of the rotation group in question. Due to TRS, all these irreducible representations are two-dimensional and can be thought of as irreducible spinor representations of $\mathbf{Z}_{2n}$, with $n$ the order of rotation. Denoting the eigenvalues of these representations by $\xi_n$ and writing the transition matrix as

$$U(\phi) = \begin{pmatrix} a(\phi) & b(\phi) \\ c(\phi) & d(\phi) \end{pmatrix}, \qquad (24)$$

the rotation symmetry requires

$$a(\phi) = a(\phi + 2\pi/n), \qquad (25)$$
$$\xi_n^2 b(\phi) = b(\phi + 2\pi/n). \qquad (26)$$

The other two entries are fixed by TRS: $c(\phi) = -\bar{b}(\phi + \pi)$ and $d(\phi) = \bar{a}(\phi + \pi)$. Upon solving these constraints, we find that for each irreducible representation, both trivial and non-trivial transition matrices are possible. This means that we can find solutions satisfying $U(\phi) = \pm U(\phi + \pi)$ with either sign, irrespective of the irreducible representation considered, and that these transition matrices may implement both trivial and non-trivial Chern numbers. The $FKM_2$ invariant is therefore still given by the parity of the Chern number for the wallpaper groups $pn$ with $n = 1, 2, 3, 4$ and 6.

Let us add a few more details concerning this argument. As before, we can decompose the $U(2)$ matrix in

$U(1)$ and $SU(2)$ parts. Owing to the rotation symmetry, the $U(1)$ part needs to satisfy $\theta(\phi) = \theta(\phi+2\pi/n)$. There is no possibility of adding a $\pi$ as was possible for TRS, because the rotation is assumed to be a unitary symmetry. There are thus no additional topological classes that arise from the $U(1)$ factor other than the two coming from TRS. The rotation symmetry also does not give new topological classes of transition matrices coming from the $SU(2)$ part, because the transition matrix is defined on a circle on which the symmetry acts transitively. The topology at fixed points was already accounted for in the representation content, so plays no role in additional topological classes of transition matrices.

## V. RELATIONS BETWEEN REPRESENTATION INVARIANTS AND TORSION INVARIANTS

That the possible existence of an $FKM_2$ invariant is independent of the irreducible representations of rotations is confirmed by Po et al. in Ref. [12]. On the other hand, the irreducible representations do not leave the Chern numbers entirely unaffected. Suppose we have a system invariant under a six-fold rotation symmetry. We could construct a TRS invariant Hamiltonian by starting with a single band with some Chern number $C$ and adding to it a band with the opposite Chern number. Depending on the actual value of this Chern number, only a particular irreducible representation appears at, say, $\Gamma$. If the Chern number is $C = 1$, the six-fold rotation acts on the bands as

$$\Gamma_1 = \begin{pmatrix} e^{i\pi/6} & 0 \\ 0 & e^{-i\pi/6} \end{pmatrix}, \tag{27}$$

whereas if $C = 3$, the irreducible representation at $\Gamma$ is $i\sigma_z$. This does not mean however that these are topologically distinct, because their $FKM_2$ invariants are equivalent. Using a similar analysis, it is also clear that one cannot create a TRS system with six-fold symmetry, starting from a single band with $C = 2$, since there is no irreducible representation that supports such a Chern number. For $p4$ and $p3$ there are also allowed and disallowed Chern numbers.

The fact that only certain Chern numbers are possible and hence only a certain number of vortices given a representation has interesting consequences. For example, for topological insulators in class $AII$ and with $p3$ symmetry, there are two possible representations, $\rho_0$ and $\rho_1$. One, $\rho_0$ can only host three vortex anti-vortex pairs (because it is a real representation) while in the other, $\rho_1$ both a single and a triple vortex anti-vortex pair is possible. In the charge 3 case, the vortices can move away from the fixed point because this does not break any symmetries nor does it close any gap. A single vortex anti-vortex pair, however, cannot move away from the fixed point because that breaks the rotation symmetry as discussed in section III of the supplementary material. This means that bands transforming under $\rho_0$ are different from those

transforming in $\rho_1$ not only because their eigenvalues are different but also $\rho_0$ can allows for a topologically distinct vortex anti-vortex configuration. This is special to space groups with a three fold rotation symmetry. For the other rotation groups of order $2n$ for $n = 0, 1, 2, 3$, an equivalent of three vortex anti-vortex pairs does not exist because that would always be an even number of vortices and thus topologically trivial.

### A. Reflection symmetry and line invariants

Reflection symmetry can be studied in a similar way. Patrametrising the transition matrix as before, and defining the action of reflection on the states as $t = i\sigma_z$, we find that $a(\phi) = a(-\phi)$ and $b(\phi) = -b(-\phi)$. It is also possible to choose a different action of reflection, $t = i\sigma_y$, which results in $a(\phi) = d(-\phi)$ and $b(\phi) = -c(-\phi)$. We will work with the latter action for convenience, but the result is independent of this choice. Trivial transition functions are then given by $a = e^{2i\phi}$ or $b = e^{2i\phi}$, whereas non-trivial ones are given by $a = ie^{i\phi}$ or $b = ie^{i\phi}$. Again we see that these are possible irrespective of the representation.

Reflection symmetries result in the presence of high-symmetry lines, which could potentially carry topological information in addition to the $FKM_2$ invariant [10]. To see this, consider transition functions along the lines $l^\perp$, which are orthogonal to the mirror plane and mapped onto themselves by TRS. These transition functions are then maps from $S^0$ to the group $M$ of matrices which act on the states. Such maps are classified by $\pi_0(M)$. On the other hand, since the lines are held fixed by the anti-unitary symmetry $Tt$, the matrices need to be real, and hence the transition functions are elements of $O(2)$, which has two disconnected components. These two components are directly related to the transition functions near a time-reversal invariant point, and have determinant $\pm 1$.

A more intuitive way of understanding such line invariants is by thinking about the Berry connection. The Berry connection is, in this case, an $SU(2)$ valued one-form on the Brillouin torus. This one-form should have correct periodicity conditions along the cycles of the torus and it should be consistent with the reflection symmetry. Let us consider a single Kramers pair. As the Berry connection is an $SU(2)$ connection, it is easiest to visualise it by projecting the connection onto the states within the pair that have the highest energy. Now consider a vortex anti-vortex pair along the $k_y = 0$ line in the BZ (vortices are fixed there due to the reflection symmetry) with the vortex at $k_x = \alpha$ and the anti-vortex at $k_x = -\alpha$. The reflection symmetry and periodicity conditions along the torus force the connection to take a special form in which all of the winding is in between the vortices, i.e. either along a line $k_x = \beta$ with $\alpha < \beta < -\alpha$ or $-\alpha < \beta < \alpha$. This winding is not really a $U(1)$ winding, but rather the two states have the topology of a Mobiüs strip along ei-

ther of these two lines. This is also in agreement with the previous discussion about homotopy groups, because the non-trivial element in $\pi_1(O(2))$ is in one-to-one correspondence with the Mobiüs strip in this situation. The integral of the Berry connection

$$\nu = \frac{1}{\pi} \int_{-\pi}^{\pi} A^I \; dk_y \qquad (28)$$

of one of the TRS channels will then be 1 mod 2, signalling the Mobiüs strip nature. As the winding is only along one of the two lines, only one of them will result in a non-trivial line invariant. Configurations in which both line invariants are 0 or both are 1 do not require the connection to have an odd number of vortex anti-vortex pairs, which again is equivalent to saying that when both line invariants have the same value, the $FKM_2$ invariant is trivial.

In the example considered in the main text, there are two parallel lines invariant under $Tt$. If both of these lines have trivial line invariants, then necessarily, the transition function $U_l$ is trivial and therefore also the transition function between the two patches in the bulk is trivial. In this case, the system as a whole is thus topologically trivial. On the other hand, if one of the two line invariants is non-trivial, the transition function, and hence the $FKM_2$ invariant also need to be non-trivial. Finally, when both line invariants are non-trivial, the $FKM_2$ invariant is trivial, since there is no non-trivial transition function between the two parallel lines.

### B. Line invariants in 3d in the presence of inversion symmetry

The line invariant is an invariant arising from choosing a non-trivial transition function along a line within the BZ. To be more precise, we cut the line in two pieces and glue them together by using this transition function. The transition function is thus a constant and not a function of a parameter. Just like with the Möbius band, this transition function is non-trivial when it has determinant minus one. When inversion symmetry is present, this cannot happen. When TRS is present, inversion symmetry in three dimensions acts on the states by either $\pm I$. The combination of TRS and inversion then acts on the states as $i\sigma_y$ and requires the transition function to satisfy

$$\sigma_y \bar{U} \sigma_y = U. \qquad (29)$$

This condition restricts $U$ to be an element of $SU(2)$ and can therefore not give rise to non-trivial topology.

### C. Non-symmorphic symmetries

Symmetries that combine point group operations with fractional lattice translations follow a similar analysis as those without the fractional translations. The only thing that is relevant is the way they act on the states. Some points in the Brillouin zone carry a different representation due to non-symmorphicity and could therefore prevent the existence of non-trivial transition functions and line invariants. For example, consider $p2gm$. At $\Gamma$ and $Y$ there is no influence of the fractional translations, but at $X$ and $M$ we have different little co-groups. The representations in which the bands can transform also changes at these points and in particular, the reflection in the $k_x$-axis constrains the transition function to be an even function of its argument. This means that at $X$ and $M$ no vortex can be present. The only allowed torsion invariants then arise from vortices at $Y$ and $\Gamma$.

## VI. K-THEORY COMPUTATIONS

The classification algorithm for crystalline insulators in class AII introduced here, is based on an intuitive picture of vortices in the Berry connection. It is expected that its results coincide with a full-fledged K-theory computation, as has been shown to be the case in class A [5]. In the case with time-reversal symmetry a proposal of how to calculate such K-theories was given in [19], but explicit computations for specific symmetry groups are missing. In a separate work, we have studied some simple examples. These will be reported in detail elsewhere, but we give a short summary of the approach here.

We first use an equivariant splitting to reduce the K-theory of tori to those of spheres. This then feeds into the Atiyah-Hirzebruch spectral sequence. The first differential gives the Bredon cohomology with coefficients in the K-theory of a point, which we take to be a twisted representation ring. One then has to compute the higher order differentials to show that either one has to go to the third page or that the spectral sequence collapses. The result is an extension problem that has to be solved in order to determine the K-theory groups that one is interested in.

We have carried out this approach for several simple such as $p2$ and $pm$. In particular, we find that in group $p2$, the calculation yields four $\mathbf{Z}_2$ invariants. This in perfect agreement with the expectation from the more intuitive approach advocated in the present paper, of counting possible topologically distinct vortex configurations.

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
