# Peer review of "Topology in time-reversal symmetric crystals"

_SciPost Physics_

## Round 2 · Referee Report · Anonymous (Referee 1) · 2018-9-8

Strengths

1. The authors present a classification of topological bands with crystal symmetry which is distinct from other works.

Weaknesses

(These points are elaborated on in the report below.)
1. The authors overstate the novelty of their own work while failing to give proper credit to earlier papers. A comparison with earlier works is lacking.
2. At different points in the paper, different definitions of topological equivalence are used.
3. It is not clear how to combine the representation and torsion invariants.
4. It is not clear whether a vortex in a band which is degenerate at points with another band is well-defined.

Report

The authors present a classification of topological bands with crystal symmetry, adapting the method they developed in Ref. 5 to the case with time-reversal symmetry. While the logic surrounding the representation invariants seems correct, the authors must give a stronger argument to show that the torsion invariants are well-defined and how they interact with the representation invariants. In addition, the authors make strong claims regarding the novelty of their work, while in fact existing classifications of topological bands with crystal symmetry have already been published; a comparison between their method and others is lacking. Thus, the following points must be addressed before the paper is suitable for publication:

1. The authors overstate the novelty of their own work while failing to give proper credit to earlier papers. The authors make the bold claim in the introduction: "The present work ... thus provides for the first time a methodical algorithm for counting topological phases in time-reversal symmetric crystals." Yet, Ref 12 earlier enumerated topological phases in time-reversal symmetric crystals. Furthermore, Bradlyn et al (Nature volume 547, pages 298–305 (2017)), which is not cited, also presents a classification for topological phases in time-reversal symmetric crystals. There are differences between each of these classifications. It is remiss to not mention the earlier papers and compare/contrast the classification schemes. This would help justify the novelty of the current paper.

Similarly, the claim, "A systematic classification of all possible topological phases in the presences of a given crystal symmetry and dimensionality, however, has
not yet been attempted" should be revised since the references mentioned above have described systematic classifications of topological phases (the methods work in any dimension, although the results are only listed in 3d.)

In addition, at the conclusion of the paper, the authors write, "We cannot yet, however, give an explicit mathematical proof that these invariants exhaust all possible topological quantum numbers." This last sentence contradicts the claim of providing the first methodical algorithm. Thus, the authors must tone down their claims to correctly describe their results.

2. The authors define two phases to be topologically distinct "if smoothly deforming
one into the other necessarily involves either closing the band gap around the Fermi level, or breaking a crystal symmetry." This definition is at odds with a classification from K-theory: two phases can be distinct by the above definition but equivalent in a K-theory classification if after adding a set of trivial bands, the two phases can be smoothly deformed into each other. Yet the authors say in the last line of the Discussion that their work could be checked by a comparison to K-theory. This contradiction is confusing: it is not clear whether the classification in the present work should, or should not, allow for the addition of trivial bands. This distinction was elucidated in Ref. 19 and has since been further explored in: Cano et al Phys. Rev. Lett. 120, 266401 (2018), Bouhon et al arXiv:1804.09719, and Bradlyn et al arXiv:1807.09729.

3. The interplay between the torsion and representation invariants is not adequately discussed. Why, in the caption to Table 1, is the total classification the direct sum of the representation and torsion invariants?

4. The authors discuss the Chern number of a single band which has a degenerate point (Kramers partner) with another band (bottom left p3: "we can still consider the Chern number of just one band within each pair".) However, the Chern number of a band is not well defined unless it is separated by an energy gap from all other bands or has a symmetry eigenvalue (for example, if spin is conserved, then each band would correspond to the opposite spin.) Thus, without spin conservation, it is not correct to think of a nontrivial Z2 invariant as describing two bands with opposite Chern number, because the Chern number of each band individually is not well-defined.

Following this logic, I am not convinced that the vortex number of a single band which is degenerate with other bands is well defined: what happens if the Berry curvature is smeared over such a large radius that it reaches the degenerate high-symmetry points?

5. In addition, could the authors clarify whether all possible representation invariants are achievable? For example, in class AI with C2 symmetry, I believe it is impossible to have a single band with an odd number of C2 eigenvalues equal to -1 (for if it were possible, then the band would have an odd Chern number (see, i.e., Eq (24) of Ref. 6), which is incompatible with time-reversal symmetry.) Have the authors checked that in class AII, there are no forbidden representation invariants?

Requested changes

In addition to the serious complaints addressed in my report, I have the following more minor comments:

1. The original paper where Kane and Mele introduced the Z2 invariant for time-reversal symmetric topological insulators should be cited when referencing the Z2 invariant ((Kane and Mele, Phys. Rev. Lett. 95, 146802 (2005)).

2. Another glaring missing citation is a reference to Teo, Fu, Kane Phys. Rev. B 78, 045426 (2011), which introduced the mirror Chern number. This is one of the few crystalline topological invariants which has actually been observed in experiment (see Hsieh, et al Nature Communications volume 3, Article number: 982 (2012) (prediction) and Tanaka, et al Nature Physics volume 8, pages 800–803 (2012) (experiment)).

3. The authors write, “In fact, it is easily seen that every combination of values for the two line invariants and one FKM invariant can be realised with precisely two distinct configurations of vortices on the high-symmetry points.” Can the authors elaborate on why this fact is “easily seen” (is the idea that each high-symmetry point with a vortex should not have a vortex in the other configuration, and vice versa?).

4. At the bottom right of p5, the authors write, “Looking at the allowed representations at Gamma, there is one real representation that allows for three vortices (or equivalently, a single charge-three vortex) to be formed there.” It would be helpful to include a table of allowed representations at Gamma in this case.

5. At the bottom left of p6, the authors write, "For example, mirror symmetries or inversions force FKM3 to be trivial.” Yet, it is known that the FKM3 invariant is not trivial with inversion symmetry, since inversion eigenvalues can be used to compute the invariant (see Ref 23). This sentence, and similar sentences in the same section, must be corrected.

---

## Round 2 · Referee Report · Anonymous (Referee 2) · 2018-9-19

Strengths

The authors present a topological classification of insulators with crystalline symmetry in Class AII using a seemingly intuitive pictures of Berry curvature vortices.

Weaknesses

While the picture provided of Berry curvature vortices does indeed lead to the right predictions of invariants as can be confirmed by K theory arguments in several cases, the argument presented here is not very rigorous. Since the occupied bands in an insulator can essentially be viewed as degenerate, the validity of a effective Hamiltonian approach for accounting for the Berry curvature at high symmetry points or elsewhere should be more adequately argued.

Report

The paper lists two classes of invariants, the representation invariants and torsion invariants. The latter are related to Berry curvature vortices and Chern numbers, and constraints on the changes in these at high symmetry and other points in the Brillouin zone are used to classify the allowed non trivial sets of Chern numbers. This approach while agreeing with those of K theory in many cases should be more adequately explained and argued.

In addition, the authors neglects to mention and compare their approach with that of earlier works which the other referee has listed.

Requested changes

While very likely correct, in its present form, its not very easy to judge the validity of the central technique of counting vortex configurations. The authors should try to make the exposition clearer. It may help to move the sections of the supplementary material in to the main text to make it more readable.

---

## Round 3 · Referee Report · Anonymous (Referee 3) · 2019-2-17

Strengths

The paper presents a novel view of topological phases in terms of vortices in the Berry curvature; this perspective leads to the discovery of a new topological invariant.

Weaknesses

The vortices and topological invariants mentioned in the paper are not rigorously defined. Therefore, it is difficult to assess the validity of the paper.

Report

I previously reviewed an earlier version of this paper. While the authors have responded to some of my comments, I am still not convinced of the validity of the manuscript. The following points need consideration:

1. Following the comment in my first report, the authors refined their definition of topologically distinct bands to exclude those related by the addition of trivial bands, writing on p1, “we start by defining two insulating phases of matter to be topologically distinct (up to the addition of trivial bands), if smoothly deforming one into the other necessarily involves either closing the band gap around the Fermi level, or breaking a crystal symmetry [5].” But with this addition, is the definition of the “representation invariants” on p2 still correct? Because the addition of trivial bands implies that the symmetry eigenvalues at high-symmetry points can change.

2. On p3 the authors write, “The pairs can be moved through the Brillouin zone, and even brought together at time-reversal invariant momenta, but they cannot annihilate there, due to the orthogonality of the electronic states within a Kramers pair. We give a more detailed analysis of this in the Supplementary Material.” But I do not see an analysis of vortices that cannot annihilate at a time-reversal-invariant momentum in the Supplementary Material. It would be nice to prove this statement explicitly.

3. As I commented in my first report, the authors write on p3, “This makes it possible to consider the Chern number of just one band within each pair, as proven rigorously in Ref. [25].” This statement is still not clear: what does it mean to define a band within a degenerate pair? (And furthermore, how do we define its Chern number?) In the one-dimensional slices in Fig 1D, it is clear that the authors intend for one band to be defined by continuously following from the lower left, for example, to the upper right. Therefore, the lower-left and lower-right portions are part of different bands. But if we rotate around the z-axis perpendicular to the plane, these lower-energy portions would be what most people would call part of the same band (the lower band.) If the authors want to define a Chern number of a band, they need to define this concept clearly.

4. I am also confused about vortices at time-reversal-invariant-momenta (TRIM). Eq (21) with m -> 0 shows that at a TRIM point, the valence and conduction bands (which are degenerate) each have a vortex in their Berry curvature of opposite sign. Since Eq (21) is generic in the case of no crystal symmetry, how is it possible for a TRIM point to not have a vortex?

5. The authors also did not address my previous question of why the classification is a direct sum: their response says it is addressed at the end of Sec C. However, the end of Sec C reads: “The total classification is the direct sum of the representation and torsion invariants. This does not exclude the possible existence of relations amongst them. … As far as the counting of topological invariants is concerned, however, the total classification is given by the sum of invariants.” This is not an explanation; it is an acknowledgement. For example, in class AII with inversion symmetry, the inversion eigenvalues completely determine the Z2 invariant. The authors acknowledge this case at the end of p6, saying that the torsion invariant is “absorbed” into the representation invariant. In this language, my initial question can be rephrased as, how do you know when the torsion invariant will be absorbed into the representation invariant and therefore the classification will not be a direct sum?

Requested changes

These are addressed in my report.

  • validity: low
  • significance: high
  • originality: good
  • clarity: poor
  • formatting: good
  • grammar: good

Author:  Jasper van Wezel  on 2019-02-21  [id 442]

(in reply to Report 1 on 2019-02-17)
Category:
answer to question

We thank the referee for reading our revised manuscript, and for the additional queries. In fact, we believe that all of the points raised by the referee are already addressed in the latest revision of our manuscript, as we discuss point by point below.

1) The referee quotes our statement on page 1, but unfortunately seems to have missed the somewhat more detailed discussion of this issue in the second paragraph of page 7. There, we explain that we "consider a trivial set of bands to be a pair with equal topological indices in which one is occupied and one unoccupied." Since the unoccupied bands are subtracted from the occupied bands when counting the number of bands with a given representation at high symmetry points, the representation invariants as defined in our manuscript are indeed invariant under the addition of such a trivial set.

2) The analysis we refer to in the text quoted by the referee are the first paragraphs of Appendix III, where we show that at TRS points, Berry curvature can be non-zero, and hence vortex-antivortex pairs do not necessarily annihilate. The referee is of course correct in asserting that this does not constitute a general proof. However, we do not feel that we need to provide such a proof, since we already refer to the (well-known and accepted) mathematical paper providing precisely the proof requested by the referee. Lemma 4.2 in reference [25], and the proof that directly follows it, shows that the Kramers pair may be written as a quaternion bundle which is well-defined even across TRS points. The absence of mixing between Kramers partners upon crossing the TRS point guaranteed by this construction also guarantees that opposite bits of Berry curvature in Kramers partners cannot annihilate at TRS points.

3) As stated in our previous reply, this construction is not our own. The possibility of relating the Z2 invariant to the Chern number of individual bands within a Kramers pair is discussed and proven in detail in reference [25], section 4.4. In particular, corollary 4.3 and theorem 4.2 provide precisely the results requested by the referee. Since this result is already available in a well-known and accepted mathematical paper which we cite, we do not think it is necessary to provide a second proof in the current manuscript.

4) As the referee correctly points out, eq 21 always leads to non-zero Berry curvature in the limit m->0. Eq. 21 thus provides an example of a local Hamiltonian close to a TRS point with non-zero Berry curvature. A TRS symmetric point without any Berry curvature cannot be described locally by eq 21. Instead, it will be described by a Hamiltonian that does not induce any band inversion across the TRS point, even though it does have a degeneracy at k=0 (in the limit m->0 of some mass parameter m). An example could be H = (k_x + k_y)\sigma_z, which does not have any curvature at k=0, but still respects TRS.

5) We thank the referee for mentioning the example of class AII with inversion symmetry. This clarifies why our statement of the invariants being a direct sum may seem counterintuitive. In the manuscript, we write about this particular example: "When inversion symmetry is present, it is known that FKM3 can be computed using the inversion eigenvalues [29], and is therefore absorbed in the representation invariants." What is meant by FKM3 being absorbed into the representation invariants, is that in the presence of inversion symmetry, the FKM3 invariant as defined in our manuscript is guaranteed to be zero. The 3D topology is then determined entirely by the representation invariants, and the classification is still a direct sum. The fact that FKM3 as defined in the manuscript turns out to be zero, even though a non-trivial topology signalled by a non-trivial product of inversion eigenvalues is possible, may indeed seem counterintuitive at first. However, our definition of the 3D invariant is the natural generalisation of FKM2 and is equivalent to that of Kaufmann, Li and Kaufmann (arxiv.org/1510.08001), which was already noted by Freed and Moore (ref [19]) to be zero in inversion symmetric systems. As Freed and Moore showed, the product of inversion eigenvalues used by Fu and Kane in ref [17] as a 3D invariant then coincides with the representation part of the topological structure. In other words: the full classification is always a direct sum of representation and "torsion" invariants, as guaranteed by K-theory (see f.e. arxiv.org/1811.02592), but care must be taken in assigning any given topological invariant to either sector, as it is not always trivial to see which invariant appears where. In the specific example of inversion symmetric 3D TI's, the invariant introduced in ref [17] appears in our classification solely as a representation invariant, while the invariant discussed in ref [19] is strictly zero.

---

## Round 3 · Referee Report · Anonymous (Referee 5) · 2019-3-22

Report

It seems there are two different languages here: one common to K-theory and one that I believe is more common to the condensed matter theory. Of course, these communities overlap and I am not sure where the line is; I myself am in the latter group. But I am now understanding the roots of some of our disagreements.

1) First, I did see the comment that trivial bands are regarded to be particle-hole symmetric. But I did not realize that the occupied bands were to be subtracted from the unoccupied bands to define the representation invariant. In fact, the current manuscript reads on p2 “We can now characterise a material with only four-fold rotation symmetry by listing the number of occupied bands for each eigenvalue at all of the high-symmetry points.” To me this contradicts the comment in the response that the representation invariant is determined by comparing the invariants in occupied and unoccupied bands: “Since the unoccupied bands are subtracted from the occupied bands when counting the number of bands with a given representation at high symmetry points…” Can the authors clarify how to compute the representation invariant and ensure that it is consistent with the addition of “trivial” bands,
using whatever definition of trivial is appropriate?

2,3) I do not have the background to read Ref. 25. (I also do not believe that paper is well known, but that is a difference of communities and semantics.) So this argument is a dead end. My only remaining comment is that the authors wrote in their resubmission letter to the editor, “In fact, the central aim of the present paper is to present a simple and intuitive picture for a topological classification of time-reversal invariant crystals that is
accessible by anyone in the field, including experimental as well theoretical physicists.” I would argue that Ref. 25 is not accessible to anyone in the field, but that the arguments in the paper do not justify computing the Chern number of overlapping bands separately. So as a reader, I am left confused, not convinced, about these statements. But that does not mean the paper is unpublishable: of course many papers in condensed matter physics rely on technical results that are outside the scope of the paper.

4) This makes sense. The example in Appendix III is quite special.

5) This seems to be a notational difference between condensed matter physics and K theory. The Fu-Kane inversion eigenvalue invariant does NOT disappear with inversion symmetry: inversion symmetry provides the index, i.e., Eq 1.2 in Ref 29 (ArXiv version). I assumed that FKM indicated the invariant in Eq 1.2 in Ref 29. This also explains the authors claim that the representation index and torsion index combine into a direct sum: if the torsion index is always defined as the part of the invariant that is not the representation index, then this is trivially true. There is an aspect of this that I don’t understand: consider a Fu-Kane-Mele type (Z2) topological insulator with inversion symmetry in class AII. Then the authors claim that the torsion indicator is trivial, so that there is not a vortex at any of the TRIM points. But when inversion symmetry is slightly broken, without closing any gaps, then the Z2 index remains nontrivial according to FKM. But also it would be impossible for a vortex to suddenly appear at a TRIM point. How is this reconciled in the author’s classification?

  • validity: -
  • significance: -
  • originality: -
  • clarity: -
  • formatting: -
  • grammar: -

Author:  Jasper van Wezel  on 2019-03-28  [id 473]

(in reply to Report 3 on 2019-03-22)

We thank the reviewer for the continued discussion and thoughtful comments. We are glad the reviewer explicitly points out they judge our paper to be publishable, and that only two small instances of insufficient clarity remain (which are only marginally related to the main content of the paper). We answer these remaining minor questions of the referee's below.

1) The definition of a "trivial band" is really not an important part of the current work. Rather, it was already implicitly introduced in some of our earlier publications, although admittedly we did not emphasize it there. For completeness, let us give a more detailed explanation here: We consider the topology of a set of bands up to a given energy E that falls within a band gap. E might be equal to the Fermi energy Ef, but does not have to be. This is similar to the usual Chern number, which can be defined for each set of bands up to an energy E, and in fact is often given for individual bands throughout the spectrum. Because E may not equal Ef, there may be both occupied and unoccupied bands below it. The representation invariants characterising the representation part of the band topology then count the number of times any given representation occurs at a given high-symmetry point in the Brillouin zone. In this counting, the occupied and unoccupied bands contribute with opposite signs. In the special case where there are only two bands, one occupied and the other unoccupied, with precisely the same representations appearing at the same high-symmetry points, their combined contribution to all representation invariants is zero. This combination of an occupied and unoccupied band with equal representation content thus constitutes the "trivial band". As pointed out by the referee, the existence of such "trivial bands" is needed to make the formal connection to K-theory. In our case, "adding a trivial band" thus amounts to adding two bands below E, one of which is occupied and one of which is unoccupied. Notice that when considering bands up to Ef (as one usually does), we should strictly speaking consider bands up to Ef plus a small delta E, so that a trivial set can be added without changing Ef. Since we consider band insulators with a finite band gap at Ef, this is always possible.

2,3,4) As the referee says, we use an established result from the literature, FKM=(-1)^C, whose proof is beyond the scope of the current paper. Unfortunately we thus cannot help the confusion of the referee's, since it is really about the result of ref 25, for which we currently do not claim to have a more accessible proof. Rather than confusing the general reader, however, we trust that our pictorial interpretation of the results of ref 25 will allow readers to use the result FKM=(-1)^C in practice, and to see how it influences the topology of crystals.

5) We are glad the confusion about the separation of representation and torsion invariants is now resolved. The question of the referee's about a very particular example is easily answered: In the presence of inversion symmetry and TRS in a three dimensional crystal, all bands are degenerate for all momentum values (since the combination TR squares to -1 but leaves k unaffected). The Berry curvature is then zero everywhere, and in our nomenclature, all topological invariants are representation invariants. As inversion is broken, the degeneracies for general k are lifted, but they do not have to be lifted in the same way everywhere. This opening of a gap for all generic k may thus constitute a band inversion across special k-points, and hence introduce Berry curvature at those high-symmetry points. Again, in our nomenclature, the situation without inversion symmetry has fewer representation invariants than the inversion-symmetric one, but gains a torsion invariant.

---

## Round 3 · Referee Report · Anonymous (Referee 4) · 2019-3-22

Weaknesses

1. The writing style of the paper can and should be significantly improved. The authors keep referring the reader to the Supplementary material for the foundations of the must crucial assertions. Since the
overall argument is heuristic rather than rigorous, the presentation becomes rather jarring. In my opinion, it would be much more helpful to present these arguments, for instance the material in Appendix III
in the main text itself.
One minor point:
Formula 1 in the text is attributed to Ref. 25, but forms of it have appeared much earlier, for instance, in Ref. 34.

Report

1. The authors claim to provide “a simple intuitive physical picture for all topological invariants with crystal symmetries.“ This claim is not however well substantiated in the paper. If it is
indeed simple and intuitive, then it should be easily explained. The picture of vortices to count the torsion invariants and correctly account for the restrictions placed by crystal symmetries seems to
reproduce existing known classifications and the authors own K-theory calculations reported elsewhere, but it should either be convincingly explained, or presented as a conjecture. In particular the statement
about the annihilation of vortices at TRS points is not convincing. The other referee has raised essentially the same objection. The authors have replied that they explain or substantiate the method in
the Supplementary material - a reading of this does not however seem to fully substantiate the claims made. Currently, it seems that the authors have developed a nice trick which gives the same answer as
more involved mathematical calculations. The authors may well be right in that the trick can perhaps be formally justified in the future - but then the authors should clearly state (in the introduction and
perhaps in the discussion at the end) that this is currently not the case. This paper is addressed to a physics audience and the mathematical rigor expected is not at the same level as for a pure
mathematics paper. The paper ends with an appendix which sketches some rigorous K-theory computations. What is missing is reasoning which has a standard of rigor which lies in between and would satisfy a physics audience. Without such reasoning, the paper is still worth publishing, but as stated before, the authors should then clarify that the paper constitutes a set of interesting observations supported by examples along with some heuristic arguments, rather than a “methodical mathematical structure which guarantees that any list of topological invariants is complete”.

  • validity: -
  • significance: -
  • originality: -
  • clarity: -
  • formatting: -
  • grammar: -

Author:  Jasper van Wezel  on 2019-03-28  [id 474]

(in reply to Report 2 on 2019-03-22)

We thank the reviewer for their reading of our manuscript and their comments.

The issues raised by the referee mainly concern the presentation of our work, rather than the content. We give an explanation for our chosen style of presentation below. Meanwhile, we are happy to note that the referee explicitly states our work to be worth publishing.

  • We choose to refer the reader to the supplementary material for all technical discussions, as noted by the referee, because the paper intends to provide both an intuitive and easily accessible pictorial understanding of band structure topology that may be applied to real materials without the need to work through all the underlying mathematics, and to provide the connection to the underlying rigorous mathematical framework for those who are interested. Although we understand the referee's assertion that the flow of presentation is somewhat impaired for advanced readers, we feel that moving material from the supplementary material to the main text, as suggested by the referee, would be an even stronger impediment to the accessibility of the work for less mathematically inclined readers.

  • The interpretation of torsion invariants in terms of Berry curvature is not a conjecture, nor something that needs to be formally justified in the future. It is already formalised and proven in for example ref. 25, and as the referee themselves points out, its essential content already appeared even earlier (for example, as pointed out by the referee, in ref 34). We do not provide an easily accessible proof of this earlier mathematical work. Nor do we claim to. As pointed out by the other referee, such a proof is beyond the scope of the current paper. We do hope, and trust, that our pictorial interpretation of the results of ref 25 will allow readers to intuitively apply the result FKM=(-1)^C in practice, and to see how it influences the topology of crystals.

---

## Round 3 · Author Response

Dear editor,

thank you very much for your email of 19 September, advising us that, based both on the published referee reports and on private communications unseen by us, you recommend resubmission of a thoroughly revised manuscript.

Having carefully studied the reports, we feel that we can adequately address all of the concerns of the referees. We appreciate the main concern of both referees, that some of the explanations in our work lack a fully rigorous mathematical backing. In fact, the central aim of the present paper is to present a simple and intuitive picture for a topological classification of time-reversal invariant crystals that is accessible by anyone in the field, including experimental as well theoretical physicists. It is for this reason that we avoided overly formal mathematical expositions in the original submission.

Although we still feel that this is the correct choice for the present work, we appreciate the advice of the referees, and urged forward by their questions, two of us (JK and JdB) set out a collaboration with pure mathematicians to study in detail the K-theoretical calculations underlying the present work in at least some of the more accessible space groups. This has now resulted in a purely mathematical paper available on the ArXiv (https://arxiv.org/abs/1811.02592), which confirms everything we write in the present submission. Because of the highly specialised and formal nature of the calculations in this follow-up work, we chose to publish it as a separate mathematics paper rather than as an appendix to the current physics paper. Nevertheless, the results of this new work strengthen our confidence in the results of the present submission, and we cite it in the revised manuscript as well as in the response to the referees below.

We are also confident that the responses below adequately address the questions and concerns raised by the referees, and that the changes to the manuscript prompted by the referee reports add to the clarity of its presentation. We are grateful for the encouraging remarks and helpful suggestions of both referees, and respectfully ask you to reconsider our submission for publication in SciPost Physics.

Yours sincerely,

Jorrit Kruthoff, Jan de Boer, and Jasper van Wezel.

Below, we respond to the remarks of the reviewers on a point-by-point basis. The relevant text from the original review reports appears between quotation marks.

Response to the First Referee

The referee starts out by stating:
"The authors overstate the novelty of their own work while failing to give proper credit to earlier papers. The authors make the bold claim in the introduction: "The present work ... thus provides for the first time a methodical algorithm for counting topological phases in time-reversal symmetric crystals." Yet, Ref 12 earlier enumerated topological phases in time-reversal symmetric crystals. Furthermore, Bradlyn et al (Nature volume 547, pages 298–305 (2017)), which is not cited, also presents a classification for topological phases in time-reversal symmetric crystals. There are differences between each of these classifications. It is remiss to not mention the earlier papers and compare/contrast the classification schemes. This would help justify the novelty of the current paper."

We thank the referee for pointing out the lack of comparison with existing literature. We agree with the referee that this is an unjust omission on our part, and include in the revised manuscript a short discussion comparing earlier approaches to ours. We now also cite the work by Bradlyn et al. mentioned by the referee.

To be specific, our work goes beyond the earlier results of Bradlyn et al. and also that of Po et al., by including a classification and intuitive discussion of topological invariants that are not related to representations of the crystal symmetry. Notice that this not a trivial addition, since although these other invariants do not arise from the crystal symmetry, both their allowed number and type may be constrained by the crystal symmetries.

A full classification of time-reversal invariant crystals must necessarily include both kinds of invariants, as presented in the current manuscript. This assertion also follows naturally from the mathematical computations in K-theory presented in https://arxiv.org/abs/1811.02592.

The referee continues:
"Similarly, the claim, "A systematic classification of all possible topological phases in the presence of a given crystal symmetry and dimensionality, however, has not yet been attempted" should be revised since the references mentioned above have described systematic classifications of topological phases (the methods work in any dimension, although the results are only listed in 3d.)"

We thank the referee for pointing out that the formulation of our claims may have been too enthusiastic. We agree with the referee that there are published works that provide a full classification of topological insulators with some particular symmetries (two-fold symmetries being a well-known example). The central purpose of our work however, is to attempt the formulation of a fully general algorithm that can describe all types of crystal symmetry, and that includes both representation invariants and topological invariants not related to crystal symmetry.

We maintain that such a fully general classification of time-reversal symmetric topological insulators has not been attempted before. For example, we are not aware of any works that classify topological phases with D4 as well as time-reversal symmetry. We do agree that partial results in this direction, within restricted settings, have been made. For example, ref. 12 (Po et al.) provides a general classification of the representation theory side. We cite these earlier works and briefly discuss how we go beyond them in the revised manuscript.

The referee adds:
"In addition, at the conclusion of the paper, the authors write, "We cannot yet, however, give an explicit mathematical proof that these invariants exhaust all possible topological quantum numbers." This last sentence contradicts the claim of providing the first methodical algorithm. Thus, the authors must tone down their claims to correctly describe their results."

We thank the referee for pointing out that the presentation of these claims was confusing. We do not want to claim that our classification is complete, although it is certainly systematic. There is no way for us to prove that there may not be more topological invariants that are not accounted for in the present approach. We explicitly include this caveat in the revised manuscript, as suggested by the referee.

The referee points out:
"The authors define two phases to be topologically distinct "if smoothly deforming one into the other necessarily involves either closing the band gap around the Fermi level, or breaking a crystal symmetry." This definition is at odds with a classification from K-theory: two phases can be distinct by the above definition but equivalent in a K-theory classification if after adding a set of trivial bands, the two phases can be smoothly deformed into each other. Yet the authors say in the last line of the Discussion that their work could be checked by a comparison to K-theory. This contradiction is confusing: it is not clear whether the classification in the present work should, or should not, allow for the addition of trivial bands. This distinction was elucidated in Ref. 19 and has since been further explored in: Cano et al Phys. Rev. Lett. 120, 266401 (2018), Bouhon et al arXiv:1804.09719, and Bradlyn et al arXiv:1807.09729."

We agree with the referee that a discussion of this point should have been included in the original manuscript. In fact, our definition of smooth deformations may be trivially rephrased in a way that does allow the addition of what would be called trivial elements in K-theory. We chose not to mention this additional structure in the original manuscript because we aimed to provide as intuitive a picture as possible. Following the remarks of the referee, we now mention the possibility of adding trivial bands in the introduction, and include a brief description of how this can be done in the discussion section.

The referee then asks a question:
"The interplay between the torsion and representation invariants is not adequately discussed. Why, in the caption to Table 1, is the total classification the direct sum of the representation and torsion invariants?"

We thank the referee for pointing out that this was not clear in our original formulation. We include a brief discussion of this point in the revised manuscript.

In short, there are two ways of thinking about the direct sum. From a physics point of view, the total classification can be understood to be a direct sum because for any representation at a fixed point, there can alway be either a non-trivial or a trivial torsion invariant. The total classification is thus the sum of the independently allowed torsion and representation invariants.

Please notice that this does not mean there cannot be any relations between the torsion and representation invariants. In fact, such relations are already known in class A, and an example within class AII is presented in section C of our submitted manuscript. We realise that it may at first sight appear confusing that the classification is always a direct sum, in spite of the possible existence of relations between torsion and representation invariants, We therefore explicitly mention this aspect at the end of section C in the revised manuscript. In section V of the supplementary materials, there is a more detailed discussion of the relations between the different types of invariants.

A second way of understanding the appearance of a direct sum, is that mathematically, K-theory is known to split into a part describing the ordinary representation theory of the point group on the one hand, and a part addressing the time-reversal and other invariants on the other. The reason for this, is that in the computation, the result always comes in the form of an exact sequence, and since the K-theory of a point is always Z, the exact sequence always splits. This is why we can write the total classification as a direct sum of representation invariants and 'the rest', which we dub torsion invariants in the present manuscript. Notice again that this distinction is made on the level of the abelian groups, and that again, it does not exclude the possible existence of relations between the integer and Z2 invariants.

The referee next raises a crucial concern:
"The authors discuss the Chern number of a single band which has a degenerate point (Kramers partner) with another band (bottom left p3: "we can still consider the Chern number of just one band within each pair".) However, the Chern number of a band is not well defined unless it is separated by an energy gap from all other bands or has a symmetry eigenvalue (for example, if spin is conserved, then each band would correspond to the opposite spin.) Thus, without spin conservation, it is not correct to think of a nontrivial Z2 invariant as describing two bands with opposite Chern number, because the Chern number of each band individually is not well-defined.
Following this logic, I am not convinced that the vortex number of a single band which is degenerate with other bands is well defined: what happens if the Berry curvature is smeared over such a large radius that it reaches the degenerate high-symmetry points?"

We sincerely regret that the justification for being able to consider the two bands within a Kramers pair separately from one another was apparently not sufficiently clearly explained in the original submission, as it is central to our approach.

In fact, this result is not our own, and has been discussed in the literature in various guises. A rigorous treatment is given for example in the mathematical paper by De Nittis and Gomi (ref. 25 in the revised manuscript). That paper argues in detail why it is possible to view the ‘Kramers’ band structure as being built from two independent bands, and how the FKM invariant is related to the Chern number of just one of the two bands within a Kramers pair.

The main point is to realise that although two bands necessarily become degenerate at time-reversal symmetric point, this does not imply mixing of time-reversal partner states. That is, starting with a single non-degenerate states at k>0, parallel transport as prescribed by the (matrix-valued) Berry connection will result in a single non-degenerate state at k<0, rather than a superposition of two non-degenerate states. This special feature of the Kramers point is what allows us (and others before us) to smear Berry curvature across a Kramers point, as suggested by the referee, and still remain without any ambiguity as to which band contains the curvature.

We appreciate the referee pointing out that the justification for treating the Chern number within just one band of a Kramers pair as equivalent to the FKM invariant was not sufficiently clearly presented in the original manuscript. We also realised that the reference given above was hidden in the supplementary material of the original submission. In the revised version, we move this reference to the main text, and include additional explanation of this important issue.

The referee then asks for clarification:
"In addition, could the authors clarify whether all possible representation invariants are achievable? For example, in class AI with C2 symmetry, I believe it is impossible to have a single band with an odd number of C2 eigenvalues equal to -1 (for if it were possible, then the band would have an odd Chern number (see, i.e., Eq (24) of Ref. 6), which is incompatible with time-reversal symmetry.) Have the authors checked that in class AII, there are no forbidden representation invariants?"

We thank the referee for this interesting question. 

In the current manuscript, we focus exclusively on class AII, and are therefore in no position to judge whether or not the specific suggestion of the referee’s pertaining to class AI is correct or not. We do know that in class AII, there are some relations between the representations and the Chern number of a single band, which we discuss in detail in section V of the supplementary material.

The referee then points out some missing references:
"The original paper where Kane and Mele introduced the Z2 invariant for time-reversal symmetric topological insulators should be cited when referencing the Z2 invariant ((Kane and Mele, Phys. Rev. Lett. 95, 146802 (2005)).
Another glaring missing citation is a reference to Teo, Fu, Kane Phys. Rev. B 78, 045426 (2011), which introduced the mirror Chern number. This is one of the few crystalline topological invariants which has actually been observed in experiment (see Hsieh, et al Nature Communications volume 3, Article number: 982 (2012) (prediction) and Tanaka, et al Nature Physics volume 8, pages 800–803 (2012) (experiment))."

We thank the referee for pointing out these omissions, and include citations to these works in the revised manuscript.

The referee continues with the observation:
"The authors write, “In fact, it is easily seen that every combination of values for the two line invariants and one FKM invariant can be realised with precisely two distinct configurations of vortices on the high-symmetry points.” Can the authors elaborate on why this fact is “easily seen” (is the idea that each high-symmetry point with a vortex should not have a vortex in the other configuration, and vice versa?)."

We thank the referee for pointing out that this statement was perhaps presumptuous. We include a revised and more detailed statement in the revised manuscript.

The referee makes a suggestion:
"At the bottom right of p5, the authors write, “Looking at the allowed representations at Gamma, there is one real representation that allows for three vortices (or equivalently, a single charge-three vortex) to be formed there.” It would be helpful to include a table of allowed representations at Gamma in this case."

We thank the referee for the constructive feedback and the suggestion. However, we believe that the inclusion of a table as suggested by the referee would only add to the already large list of figures and tables in the manuscript without adding much in terms of content. Moreover, the representation theory of this group (and others) can nowadays be easily found on the Bilbao crystallographic server. We therefore prefer not to include such tables in the present manuscript.

The referee finishes with:
"At the bottom left of p6, the authors write, "For example, mirror symmetries or inversions force FKM3 to be trivial.” Yet, it is known that the FKM3 invariant is not trivial with inversion symmetry, since inversion eigenvalues can be used to compute the invariant (see Ref 23). This sentence, and similar sentences in the same section, must be corrected."

We thank the referee for pointing out this imprecision. We correct this and similar sentences in the revised version of the manuscript.

Response to the Second Referee

The referee starts out by stating:
"While the picture provided of Berry curvature vortices does indeed lead to the right predictions of invariants as can be confirmed by K-theory arguments in several cases, the argument presented here is not very rigorous. Since the occupied bands in an insulator can essentially be viewed as degenerate, the validity of an effective Hamiltonian approach for accounting for the Berry curvature at high symmetry points or elsewhere should be more adequately argued."

We thank the referee for the constructive remarks concerning this point. We agree with the referee that the work presented in the current manuscript does not include rigorous mathematical proofs. However, as mentioned before in the reply to the editor, the explicit aim of the present paper is to present a simple and intuitive picture for a topological classification of time-reversal invariant crystals that is accessible by anyone in the field, including experimental as well theoretical physicists. It is for this reason that we purposefully avoided formal mathematical expositions in the original submission.

To pursue the kind of mathematical rigour suggested by the referee, requires the computation of the K-theories associated with the topological phases we discuss in the manuscript, which would inevitably turn the paper into a formal mathematical treatise, rather than a physics article. We give some more formal arguments to back up our results in the supplementary material, but prefer not to put them in the main text.

Although we still feel that this is the correct choice for the present work, we appreciate the advice of the referee, and urged forward by their questions, two of us (JK and JdB) set out a collaboration with pure mathematicians to explicitly compute the K-theory groups involved in some of the cases where the current manuscript suggests new topological invariants. This work is now available as a purely mathematical paper on the ArXiv (https://arxiv.org/abs/1811.02592), and confirms the classification as proposed in the current manuscript in all cases for which we have explicit results in K-theory. Of course, this is still not a complete proof of our vortex picture, but it gives substantial evidence in that direction, and strengthens our confidence in the results of the present submission.

The referee continues by stating:
"The paper lists two classes of invariants, the representation invariants and torsion invariants. The latter are related to Berry curvature vortices and Chern numbers, and constraints on the changes in these at high symmetry and other points in the Brillouin zone are used to classify the allowed non trivial sets of Chern numbers. This approach while agreeing with those of K theory in many cases should be more adequately explained and argued.
(…)
While very likely correct, in its present form, its not very easy to judge the validity of the central technique of counting vortex configurations. The authors should try to make the exposition clearer. It may help to move the sections of the supplementary material in to the main text to make it more readable."

We regret that the referee did not find our exposition sufficiently clear.
Since the referee does not give concrete examples of where our explanation lacks clarity, or what they would consider to be sufficiently clear explanations, we cannot respond directly to this comment. However, we improved the overall clarity of the manuscript by adding additional explanations throughout the manuscript. As suggested by the referee, we also moved some references and explanations from the supplementary material to the main text. As pointed out also by the first referee, especially ref. 25 by De Nittis and Gomi is an important mathematical result that is central to the understanding of why the ‘Kramers’ band structure may be viewed as consisting of two separate bands, and how the FKM is related to the Chern number of just one of these two bands.

Finally, the referee remarks:
"In addition, the authors neglects to mention and compare their approach with that of earlier works which the other referee has listed."

We thank the referee for pointing out this omission, and include a comparison to and discussion of earlier works in the revised manuscript.

---

## Round 3 · List of Changes

All changes are mentioned in the response to the referees above.

---

## Editorial Decision

rejected_or_withdrawn